# Microcavity-like exciton-polaritons can be the primary photoexcitation in bare organic semiconductors

Raj Pandya[1], Richard Y. S. Chen[1], Qifei Gu[1], Jooyoung Sung[1], Christoph Schnedermann[1], Oluwafemi S. Ojambati[1], Rohit Chikkaraddy[1], Jeffrey Gorman[1], Gianni Jacucci[2], Olimpia D. Onelli[2], Tom Willhammar[3], Duncan N. Johnstone[4], Sean M. Collins[4], Paul A. Midgley[4], Florian Auras[1], Tomi Baikie[1], Rahul Jayaprakash[5], Fabrice Mathevet[6], Richard Soucek[7], Matthew Du[8], Antonios M. Alvertis[1], Arjun Ashoka[1], Silvia Vignolini[2], David G. Lidzey[5], Jeremy J. Baumberg[1], Richard H. Friend[1], Thierry Barisien[7], Laurent Legrand[7], Alex W. Chin[7], Joel Yuen-Zhou[8], Semion K. Saikin[9,10], Philipp Kukura[11], Andrew J. Musser[12] & Akshay Rao[1]✉

Strong-coupling between excitons and confined photonic modes can lead to the formation of new quasi-particles termed exciton-polaritons which can display a range of interesting properties such as super-fluidity, ultrafast transport and Bose-Einstein condensation. Strong-coupling typically occurs when an excitonic material is confided in a dielectric or plasmonic microcavity. Here, we show polaritons can form at room temperature in a range of chemically diverse, organic semiconductor thin films, despite the absence of an external cavity. We find evidence of strong light-matter coupling via angle-dependent peak splittings in the reflectivity spectra of the materials and emission from collective polariton states. We additionally show exciton-polaritons are the primary photoexcitation in these organic materials by directly imaging their ultrafast ($5 \times 10^6$ m s$^{-1}$), ultralong (~270 nm) transport. These results open-up new fundamental physics and could enable a new generation of organic optoelectronic and light harvesting devices based on cavity-free exciton-polaritons

[1] Cavendish Laboratory, University of Cambridge, J.J. Thomson Avenue, CB3 0HE Cambridge, UK. [2] Department of Chemistry, University of Cambridge, Lensfield Road, Cambridge CB2 1EW, UK. [3] Department of Materials and Environmental Chemistry, Stockholm University, Stockholm, Sweden. [4] Department of Materials Science and Metallurgy, University of Cambridge, 27 Charles Babbage Road, CB3 0FS Cambridge, UK. [5] Department of Physics & Astronomy, University of Sheffield, S3 7RH Sheffield, UK. [6] Institut Parisien de Chimie Moléculaire (IPCM), Sorbonne Université, 4 Place Jussieu, 75005 Paris, France. [7] Institut des NanoSciences de Paris (INSP), Sorbonne Université, 4 place Jussieu, 75005 Paris, France. [8] Department of Chemistry and Biochemistry, University of California San Diego, La Jolla, CA 92093, USA. [9] Department of Chemistry and Chemical Biology, Harvard University, 12 Oxford Street, Cambridge, MA 02138, USA. [10] Kebotix Inc., 501 Massachusetts Avenue, Cambridge, MA 02139, USA. [11] Physical and Theoretical Chemistry Laboratory, Department of Chemistry, University of Oxford, South Parks Road, Oxford OX1 3QZ, UK. [12] Department of Chemistry and Chemical Biology, Cornell University, Baker Laboratory, Ithaca, NY 14853, USA. ✉email: ar525@cam.ac.uk

Exciton-polaritons (EPs) are quasiparticles formed by the hybridisation of excitons with light modes. As organic semiconductors sustain stable excitons at room-temperature, these materials are being actively studied for room temperature polaritonic devices[1–3]. This is typically in the form of cavity-based systems, where molecules are confined between metallic or dielectric mirrors[4–6] or in a plasmonic gap[7,8]. In such systems strong light-matter coupling gives rise to polariton splittings on the order of 200 to 300 meV[6]. A wide range of phenomena have been demonstrated in cavity-polariton systems including super-fluidity[9], precisely controlled chemical reactions[3] and long-range energy propagation[10]. Here, using a range of chemically diverse model organic systems we show that interactions between excitons and moderately confined photonic states within the (thin) film can lead to the formation of EPs, with a defined lifetime, even in the absence of external cavities. We demonstrate the presence of EPs via angular dependent splittings in reflectivity spectra on the order of 30 meV and collective emission from ~$5 \times 10^7$ coupled molecules. Additionally, we show that at room temperature these EPs can transport energy up to ~270 nm at velocities of ~$5 \times 10^6$ m s$^{-1}$. This propagation velocity and distance is sensitive to, and can be tuned by, the refractive index of the external environment. However, although sensitive to the nanoscale morphology the formation of the exciton-polariton states is a general phenomenon, independent of underlying materials chemistry, with the principal material requirements being a high oscillator strength per unit volume and low disorder. These results and design rules will enable the harnessing of EP effects for new applications in optoelectronics, light harvesting[9,11,12] and polariton mediated chemistry without the limiting requirement of an external cavity.

Early discussions of EPs in organic materials focussed on the concept of 'intrinsic' or 'bulk' polaritons, where EPs are treated as modes of an electromagnetic wave propagating in matter[13–15]. When the frequency of incident light is in the range of the excitation resonance frequencies in some, often highly absorbing materials, the oscillating polarisation emitted by that excitation will interfere with the incident wave, resulting in neither the photon nor exciton being good eigenstates[16,17]. The resulting quanta of mixed polarisation and electromagnetic waves are called polaritons, and we further distinguish these as 'intrinsic'. Intrinsic polaritons show anomalous dispersion and can be identified through coherent oscillations patterns in the retardation of propagating ultrafast pulses[18–22]. In organic materials this formalism effectively rationalises the metallic reflectivity of some macroscopic organic single crystals and has been investigated in detail elsewhere[23–27]. However, to the best of our knowledge, such 'intrinsic' polaritons have neither a well-defined photonic state nor distinct EP lifetime in organic systems, and it is unclear to what extent they carry electronic population; in inorganic systems they may support a lifetime[28]. Furthermore, the organic material properties that govern these 'intrinsic' polariton effects – aside from strong absorption – are not well known, and their apparent dependence on bulk material makes them difficult to apply in devices. The coupling of an exciton and a waveguide mode in an optical resonator gives rise to a second type of EP[29–31]. Here, two flat surfaces, typically facets of a 1D nanostructure, act as a high quality factor Fabry-Pérot resonator[32]. These systems show EP lasing[33], but because only photonic information is propagated, they have not been widely developed for optoelectronics such as LEDs or photovoltaics. A similar situation holds for the case of excitons coupled to Bloch surface waves (Bloch-wave polaritons), where ultralong range transport of predominantly light-like states has been achieved, but not matter-like states crucial for many devices[34,35].

A principal requirement for EP formation is significant light-matter coupling, either via a high material oscillator strength or the precise confinement of optical fields (e.g. cavities and nanophotonic systems), with ideally a combination of both[8,36,37]. Hence recent studies of EPs have largely focussed on cavity-based systems, where molecules are confined between metallic or dielectric mirrors[4–6] or in a plasmonic gap[7,8]. A wide range of phenomena have been demonstrated with cavity-polaritons including super-fluidity[9], controlled chemical reactions[3] and long-range energy propagation[10]. However, the need for a highly reflective external structure to confine the light field, makes sample fabrication and practical application to optoelectronic devices complex with issues of charge injection and light out-coupling. Consequently, to truly realise the properties of EPs for future optoelectronic device applications, such as long-range energy transport, coherent emission, etc, what is required are materials where there is an interaction between an exciton and an intrinsically moderately confined photonic state, but no external cavity. Such a moderately defined photonic component would endow the EP state with defined lifetime, i.e. a state with a distinct and potentially observable linewidth, without requiring an external cavity. Here, we study model organic conjugated polymer, J-aggregate and H-aggregate systems. While the primary photoexcitation in organic materials have long been considered to be excitons, we provide evidence in the form of angle-dependent splitting in reflection spectra, Rabi flopping of emission intensities and ultrafast, tunable, long-range energy transport, that these systems support EP in the absence of an external cavity, which are in-fact the primary/initial photoexcitation. We demonstrate these EP effects to transcend the underlying molecular building blocks but to be connected to the nanoscale morphology of samples, providing a set of design rules to realise such phenomena in other organic materials.

## Results and discussion

Figure 1a–c depicts molecular structures, packing motifs and electron microscopy images of the three model systems: poly-diacetylene (PDA), a conjugated polymer[38,39], in the form of aligned 2–5 µm chains; PIC, a molecular semiconductor that can self-assemble to form nanotubular J-aggregates based on the brickwork packing of cyanine dyes[40–42], with length 1–5 µm and tube diameter of 2–5 nm; and a perylene dimiide (PDI) which assembles to form ~100 µm long 'nanobelts' consisting of H-aggregated PDI molecules[43,44]. In each case we fabricated films of the samples (see Supplementary Methods for film fabrication details) where: PDA chains are dilutely and homogeneously aligned in a matrix of their monomer (~100 nm interchain separation); PIC nanotubes are suspended in a sucrose-trehalose matrix (~5 nm separation between tubes with some bundling); and PDI nanobelts are dispersed on a glass substrate (~100 nm separation with some bundling, however unless otherwise stated individual and non-overlapping nanobelts are measured). The systems have disparate chemical building blocks and packing motifs but all of them have high oscillator strengths and intrinsically low disorder (see structural characterisation in Supplementary Note 1).

In Fig. 1d–f we show the results of microscopic reflectivity measurements (sample illumination/collection area ~3 µm$^2$; Supplementary Note 2) for PDA and PIC, where in each material we see the strong absorption transition splits into two peaks in the reflectivity spectra. In PDA the zero-phonon transition (1.965 eV) is split by typically $17.5 \pm 0.5$ meV, whereas in PIC the J-aggregate exciton transition (2.132 eV) is split by $39.6 \pm 0.6$ meV. Additionally, in PIC we find this feature has an angle dependence distinct from any intrinsic Fabry-Pérot modes that can also be detected by such measurements, revealing clearly the state has partial photonic character (see Fig. 1f and Supplementary Note 3).

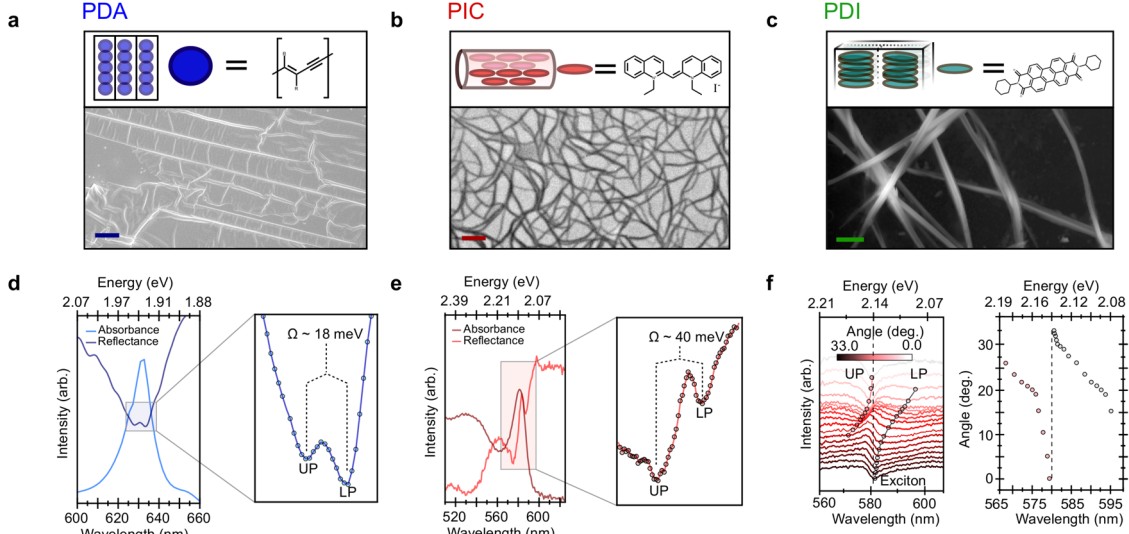

**Fig. 1 Structural characterisation of quasi-one dimensional organic semiconductors studied and experimental evidence for light-matter coupling. a** Cartoon and scanning electron-microscopy (SEM) image of PDA. The polymer chains consist of a single-double-single-triple bonded back bone and are near perfectly aligned in 100 $\mu m^2$ domains. Scale bar is 10 $\mu m$. **b** Molecular packing and transmission electron-microscopy image of PIC nanotubes. The cyanine based monomers pack in a brickwork pattern to form nanotubes. They are embedded in a rigid sucrose-trehalose matrix and tend to be highly bundled in films; scale bar 100 nm. **c** PDI molecules pi-face stack in a quasi-H-aggregate fashion and in four distinct blocks (black outlines) to form ~100 $\mu m$ long, 50 nm wide nanobelts; scale bar 250 nm. **d** Absorption (light blue) and specular reflection (dark blue) spectra of PDA. Zooming into the zero-phonon peak at 1.965 eV (right) shows it is split by ~18 meV ($\Omega$) into two branches, an upper (UP) and lower polariton (LP). **e** Absorption (maroon) and specular reflection (red) spectra of PIC. The excitonic peak at 2.132 eV is split by ~40 meV due to the formation of exciton-polaritons (right). **f** Angle resolved reflectivity for PIC. Solid red and black lines highlight angle-dependent dips below and above the excitonic transition (dashed line), respectively. The energy of the dips as a function of angle (adjacent) reveals the typical anti-crossing behaviour of strong coupling between a cavity photon-mode and an excitonic transition (dashed line).

Specular reflection measurements are commonly used to identify the formation of exciton-polaritons (EPs), where the coupling of light and exciton absorption at the exciton resonance results in the splitting in the reflectivity spectrum into upper and lower polariton bands[45]. The specular reflection hence suggests that these films support EPs, despite the absence of any cavities or nanophotonic systems. The angle-dependent measurements which allow us to distinguish lower and upper EP states, provide further strong proof of EP formation[35]. We hence propose that the exciton states in the organics are admixed with mildly confined photonic modes supported by the dielectric constant mismatch between the molecules and the surrounding environment.

Measuring the reflectivity at ~70 different sample locations in PDA and PIC (Supplementary Note 2), demonstrates the magnitude of the splitting does not vary significantly from site-to-site (within our resolution), but is only resolvable in ~30% of the regions examined. We attribute these variations to nanoscale inhomogeneities, e.g. from sample thickness or bundling of nanotubes (Supplementary Note 1 and 3), which give rise to regions of increased oscillator strength. Transfer matrix modelling suggests that only regions with such locally enhanced oscillator strength can give rise to the splittings we detect. At the same time, our modelling suggests it is the thickness variations which determine how resonant the weakly trapped optical modes are with the exciton. This provides further evidence that our observations are distinct from the aforementioned 'intrinsic' polaritons[27], which have been almost exclusively observed in highly polished and reflective organic single crystals[15,46]. Furthermore, these results also suggest that disorder in thickness directly impacts the position of the cavity mode to a greater degree than in well-controlled Fabry-Pérot cavities. As a result, it is likely only a localised subset of tubes/chains within our

microscope spot contribute to the splitting. Consequently, the inhomogeneous broadening in the room-temperature absorption spectrum exceeds the splitting we can clearly observe in reflectivity. The model also shows that the splitting scales with the square root of absorbance, as expected for collective light-matter coupling and confirming the assignment to an exciton-polariton (see Supplementary Note 3 for further discussion and modelling). These observations are akin to the interaction between excitons and moderately confined photonic states in cavity-based systems, with the UP and LP corresponding to the excitation of polaritons. According to our computational simulations, we estimate (Supplementary Note 4) that the aggregates occupy as much as 10–30% of the mode volumes of these photonic modes, leading to modest collective light-matter couplings (see later) that create EPs separated by distinct Rabi splittings (Fig. 1d, e). Hence, in contrast to molecular EP systems in cavities, there is no significant energetic separation between EPs and the dark states (i.e., exciton states which, in the absence of disorder, do not admix with photons). The latter, which we hereafter call subradiant, are consequently endowed with small fractions of photonic character.

While static reflectivity measurements provide direct evidence of EP states in PDA and PIC, in PDI this is precluded by a multitude of homogeneously broadened vibronic peaks. The lack of the characteristic peak splitting may also be attributed to the absence of confined photon modes that are resonant with the highly absorbing molecular transitions (Supplementary Note 4). We track instead the intensity of emitted photoluminescence (PL) as a function of driving optical field (laser fluence), at room temperature. In the presence of light-matter coupling an oscillatory light-matter response is expected due to periodic exchange of energy between the 'ground + photon' and 'excitation + no photon' states, driven at the Rabi frequency, $\widetilde{\Omega}_R = \mu_{12}|E_0|/\hbar$ ($\mu_{12}$

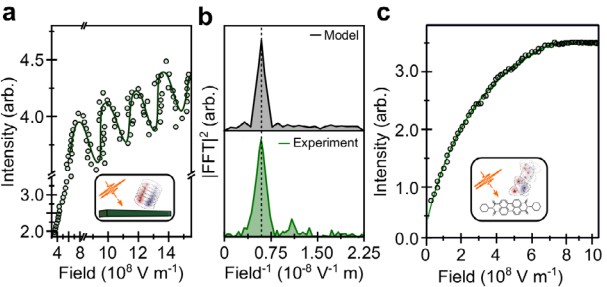

**Fig. 2 Light-matter coupling in PDI nanobelts. a** Emission intensity as a function of optical field (laser fluence) for PDI nanobelts. After increasing linearly, the PL intensity then begins to oscillate at a frequency of 0.71 $V^{-1}$ m after 8 ×$10^8$ V $m^{-1}$ revealing coherent emission from ~5 ×$10^7$ molecules (inset). Data is obtained from averaging 3 up/down power sweeps. Solid black line show fit of model to data (Supplementary Note 5). In a similar manner to the reflectivity measurements in Fig. 1c,d not all sample locations exhibit coherent emission, likely due to nanoscale ordering beyond the resolution of our experiment. **b** Modelling the data[51] allows extraction of the damping time for Rabi flopping (Supplementary Note 5). Strong agreement between the model and experiment is highlighted by the Fourier transform of the oscillation (bottom). The extra frequencies in the experimental data arise from the detuning of the laser pulse (590 nm, FWHM ~10 nm) from the excitonic transition ($\lambda_{max}$ = 512 nm). **c** In the case of the individual PDI chromophores (or densely packed nanobelts, Supplementary Note 5) no such behaviour is observed (solid line shows model fit) with the PL intensity increasing monotonically with optical field before saturating.

is the transition dipole moment and $|E_0|$ is the magnitude of the driving optical electric field)[47–50]. For a given pulse duration, Rabi flopping is expected in the power dependence of the emission due to changing $\tilde{\Omega}_R$ (here from 2.6 THz to 4.4 THz) with the amplitude of the incident field $E_0$. As shown in Fig. 2a, for isolated PDI wires, the PL intensity initially increases linearly with the applied field before beginning to oscillate at high optical fields (these fluences are similar to those used in femtosecond transient absorption microscopy experiments discussed later). Modelling the oscillation frequency[51] (Fig. 2b), yields a damping time of 50 ± 10 fs i.e. time for which well-defined superpositions of the ground and excited states exist. Although pump-induced oscillations are not necessarily indicative of strong-coupling[48], in the case of PDI wires we suggest that the formation of polaritons is required to overcome any rapid competing decay processes such as annihilation or trapping[47]. This in turn leads to cooperative interaction and emission from the densely packed molecules that make up the nanobelt (~5 ×$10^7$ within our collection volume). Such behaviour is akin to resonance fluorescence observed in two-level atomic systems and typically only occurs with single molecules in high $q$-factor cavities[52]. Importantly such oscillatory behaviour is not observed in isolated monomers in solution or disordered bundles of PDI wires (Fig. 2c and Supplementary Note 5).

To explore the use of cavity free EPs for light harvesting and energy transport we study their spatiotemporal dynamics using femtosecond transient absorption microscopy (fs-TAM) as shown in Fig. 3a. Here, a broadband sub-10 fs pump pulse (520 – 650 nm) is focussed to the diffraction limit (FWHM ~270 nm) by a high numerical aperture (~1.1) objective onto the sample, with a wide field, counter-propagating probe pulse (8 fs, 680 – 790 nm, FWHM ~15 μm) focussed with a concave mirror, used to monitor the normalised change in image transmission with pump on ($T_{on}$) and off ($T_{off}$) as function of time delay between the pulses

($\frac{\triangle T}{T} = \frac{T_{on} - T_{off}}{T_{off}}$). By subtracting the extent of the spatial profile at a time $t$, from that at zero time delay between pump and probe ($t_0$) the propagation of population can be monitored (further details see Supplementary Note 6)[53–56]. The limit to localisation precision is determined by how well different spatial profiles can be resolved (right panel Fig. 3a); based on the signal to noise ratio of the measurement this typically amounts to ~10 nm. Although, this method allows high-resolution of the spatial transport distance, we emphasise that our imaging of spatial features on the films is still limited to the diffraction limit[53,57]. Measurements were carried out on ~40 locations across multiple films per system, resulting in an unprecedented statistical picture of how transport relates to morphology

Representative fs-TAM images obtained for PDA (blue), PIC (red) and PDI (green) following photoexcitation are shown in Fig. 3b–d. We probe the energy transport by monitoring the stimulated emission (SE) bands of PDA (670 nm) and PIC (600 nm), and the photoinduced absorption (PIA) of PDI (720 nm). These wavelength assignments are based on standard measurements of thin films of these materials without EPs[43,58,59]. The reported behaviour (below) is independent of the exact probe wavelength within the band (Supplementary Note 7) and it can be taken on the timescales studied here we simply track the spatial spread of the transient absorption response. In all three cases, there is a large and rapid expansion of the initial excitation spot. The 1D nature of the samples means that the growth is highly anisotropic with the excitation spot being a near symmetric Gaussian at $t_0$ but rapidly elongating in the wire direction (in the case of PDA and PIC this will be the average of wires within the pump spot, in PDI single wires are excited; see Supplementary Note 1). To quantify the extent of spatial transport, we extract the FWHM of each image along the propagation axis and convert it to a Gaussian standard deviation σ (see Supplementary Note 6 includes discussion of propagation in orthogonal direction). The $t_0$ frame is identified from the σ value expected based on the diffraction limit of our pulses (σ ~135 nm corresponding to a FWHM ~310 nm) and by fitting the rise of the signal with the instrument response (Supplementary Notes 6 and 7). In PDA the expansion is largest with σ increasing from roughly the diffraction limit of 134 ± 5 nm at t = 0 fs to 336 ± 5 nm at t = 200 ± 3 fs. For PIC σ increases from 143 ± 5 nm to 231 ± 3 nm in the same time period and in PDI the expansion is smallest, with a σ of 149 ± 8 nm at $t_0$ and σ of 194 ± 6 nm at t = 200 ± 3 fs. The large initial (sub-200 fs) expansion in all three samples suggests that excitations can travel far beyond the sub-5 nm range reported on these timescales for FRET mediated systems[60,61].

To analyse the data further we plot the mean square displacement $MSD = \sigma(t)^2 - \sigma(t_0)^2$ as a function of time in Fig. 4a–c. The behaviour appears biphasic in nature for all three samples, with an initial ultrafast sub-50 fs phase labelled $R_1$ (time resolution ~10 fs), followed by a slower expansion over the following 200 fs, labelled $R_2$. In the first phase $R_1$, before scattering and diffusive behaviour dominate, we can decompose our initial excitation's time evolution into two Gaussian subpopulations (for the 1D motion we observe, see Supplementary Note 8 for further details), with the overall dynamics modelled as,

$$\sigma(v_g, t) = \sigma_0 + \exp(-t/\tau)\frac{v_g^2 t^2}{\sigma_0}. \quad (1)$$

The time constant $\tau$ captures the characteristic scattering time of the population and $v_g$ the ballistic velocity in this early time regime of coherent transport. Fitting Eq. 1 to the $R_1$ region in Fig. 4a–c, allows for transport velocities for all three systems to be determined in the range of $2.0 \times 10^6$–$5.8 \times 10^6$ (± 0.03) m $s^{-1}$ (PDI < PIC < PDA), with scattering times of 14 – 70 (± 5) fs and

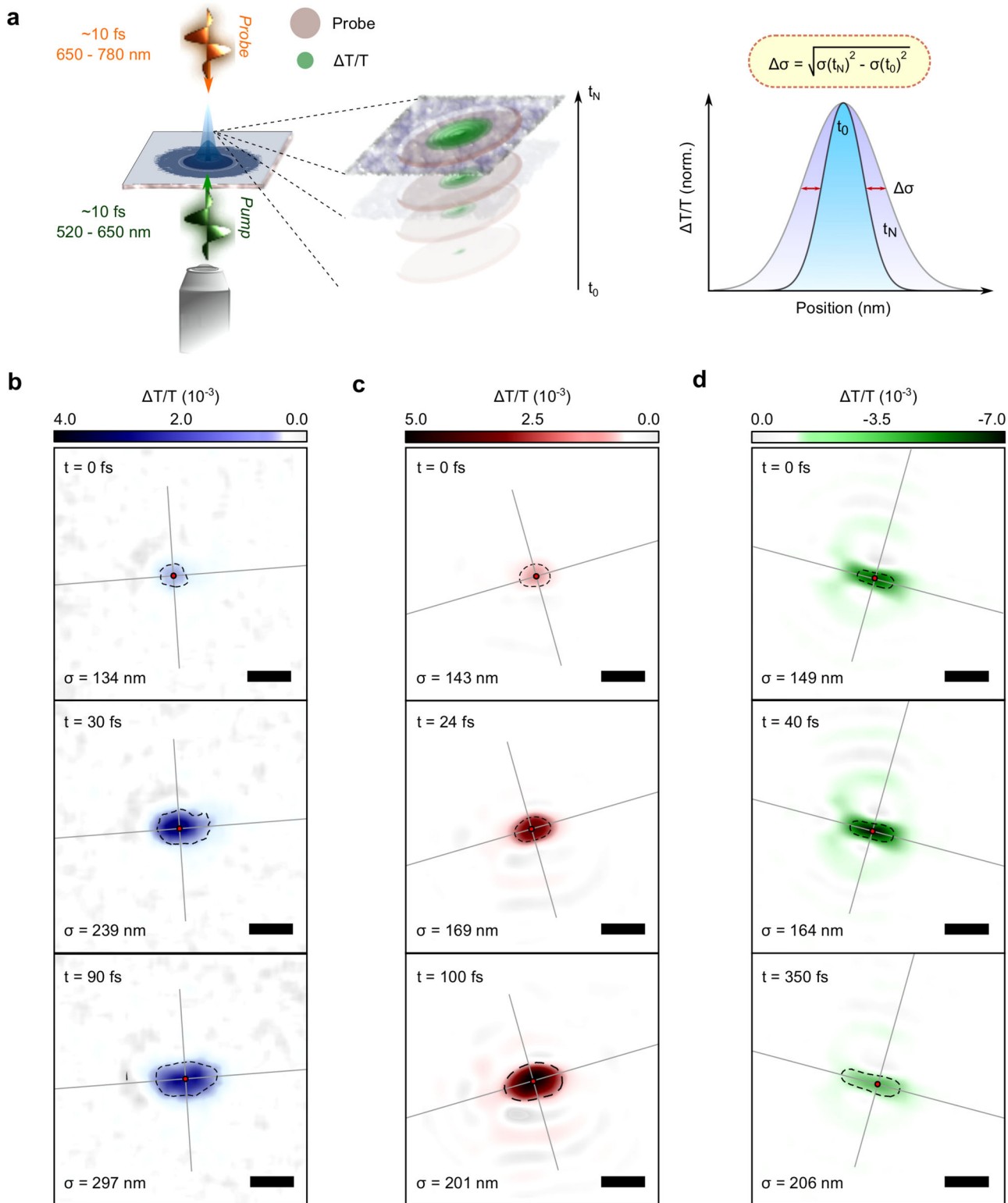

**Fig. 3 Femtosecond transient absorption microscopy of organic nanostructures. a** Schematic of fs-TAM. A diffraction limited pump pulse (green) is focused onto the sample by a high numerical aperture microscope objective. A probe pulse (yellow) is focussed from the top onto the sample in the wide-field. The transmitted probe is collected by the objective and imaged onto a digital camera (Supplementary Methods). Comparison of the spatial extent of the signal (stack) at different time delays allows us to dynamically track changes in the population with ~10 nm precision (right hand graph). **b–d** fs-TAM images of PDA, PIC and PDI at selected time delays following photoexcitation. The pump pulse covers the entire (excitonic) absorption of the systems, with probing carried out at 670 nm, 600 nm and 720 nm respectively. In all images in Fig. 1e–g the scale bar is 500 nm and the dotted line indicates the radial Gaussian standard deviation ($\sigma$; numerical value bottom left) from the excitation centre of mass (red circle). Grey lines indicate the principle transport axes, along and orthogonal to the average wire direction (see Supplementary Note 6 for respective line cuts).

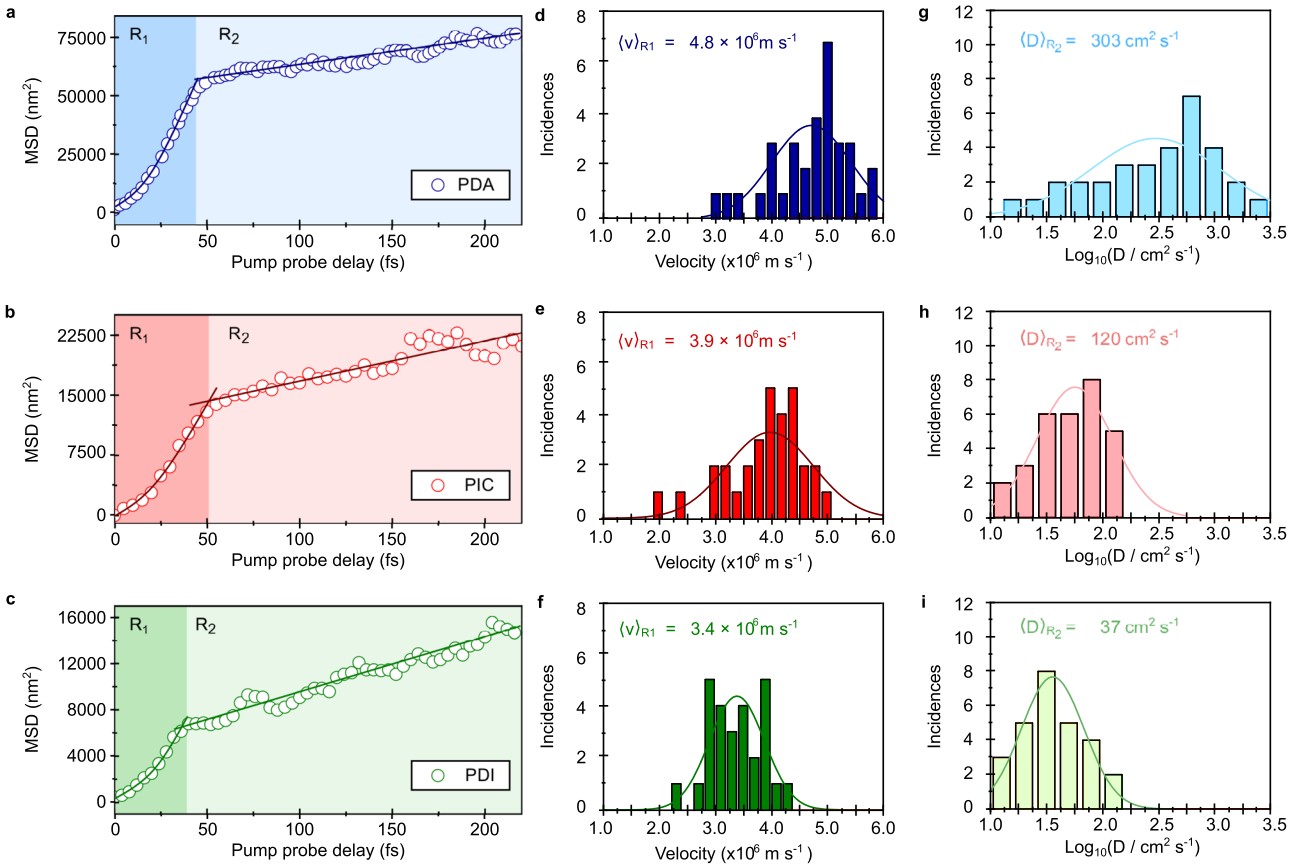

**Fig. 4 Dynamics of energy propagation and distribution in transport lengths. a–c** Representative plots of MSD ($\sigma(t)^2 - \sigma(t_0)^2$) as a function of time for PDA (**a** blue), PIC (**b** red) and PDI (**c** green). The transport is divided into two regimes $R_1$ and $R_2$. Solid lines show a fit to the two regions (Equation 3 and $Dt^\alpha$) from which a transport velocity (v) and diffusion constant (D) can be estimated. **d–i** fs-TAM measurements are repeated on many different sample locations. Histograms show the frequency of velocity and $Log_{10}(D)$ (logarithm of diffusion coefficient, D) for PDA, PIC and PDI respectively. The data is divided between $R_1$ (dark shading) where a velocity best describes the transport and $R_2$ (light shading) where a diffusion coefficient is most appropriate for characterising the transport. Solid line shows a normal distribution fit to the data. The mean transport parameters obtained in both phases are indicated for comparison.

transport distances between 50–200 nm. The speed of transport is two-orders of magnitude larger than that found in organic semiconductors[62–64], one order above the maximum velocity previously observed in PDAs[65] and above that reported for excitons in low temperature epitaxial GaAs[66] and free electrons in metallic films[67]. Repeating fs-TAM measurements on many different sample locations allows us to build up a histogram of transport velocities, shown in Fig. 4d–f. The relatively small spread of velocities in $R_1$ suggests that population transport in this regime is related to an intrinsic property of the material and is not greatly perturbed by differences in nanoscale/local ordering and morphology that may be present in different sample locations within the sample. Fitting Eq. 1 to the full MSD profile (Supplementary Note 8) shows that it strongly deviates in $R_2$. Consequently, we fit the MSD with an equation $MSD = Dt^\alpha$ ($t$ is propagation time in $R_2$, $\alpha$ is a coefficient relating to the nature of transport, $\alpha = 1$ diffusive transport $\alpha < 1$ sub-diffusive transport, $\alpha > 1$ ballistic transport). This equation, the solution to the classic differential equation from Brownian diffusion equation in the absence of exciton-exciton annihilation (as we observe, see Supplementary Note 9), is commonly used to characterise transport in organic semiconductors and strictly relates to incoherent transport. For all three samples we find $\alpha = 1$ and in Fig. 4g–i plot the diffusion coefficient for each of the sample locations. The average diffusion coefficient ($\langle D \rangle$) follows the same trend PDA > PIC > PDI: $\langle D \rangle_{R_2} = 303 \pm 25$ cm$^2$ s$^{-1}$ (PDA), $120 \pm 10$ cm$^2$ s$^{-1}$ (PIC) and $37 \pm 10$ cm$^2$ s$^{-1}$ (PDI). The spread in $D$

(and $v$) in $R_2$ is significantly larger than in $R_1$ (ratio of standard deviation for velocity ($R_1$) and D ($R_2$) of PDA = 4.1; PIC = 2.3; PDI = 1.9). This would suggest that local morphology and disorder potentially has a greater effect in this regime and may also explain the above ordering trends (see Supplementary Note 6 for further discussion and errors on value). However, we observe no correlation between the decay lifetime of the electronic signal and the diffusion coefficient (Supplementary Note 10); this suggests in the early-time regimes the electronic lifetime is not necessarily a good metric for the diffusion coefficient. We note the velocities and log-diffusion coefficients on Fig. 4d–i show a log-normal distribution. This reflects the random selection of sample locations for measurement and distribution of local disorder within the self-assembled films. In summary, energy transport in all three materials can be characterised by an initial period of ballistic coherent motion, followed by scattering into an incoherent diffusive regime.

To understand the origins of this transport behaviour, we model the electronic structure of all three model systems (see Supplementary Note 11 for details) and obtain the maximum ballistic velocities based on the band dispersion for excitons in each system: $v \approx 0.2 \times 10^6 \pm 0.06$ m s$^{-1}$ for PDI; $v \approx 0.12 \times 10^6 \pm 0.04$ m s$^{-1}$ for PIC; and $v \approx 0.3 \times 10^6 \pm 0.01$ m s$^{-1}$ for PDA. These velocities are one order of magnitude lower than those obtained experimentally in the $R_1$ region, showing that they cannot arise from purely excitonic transport. This one order of magnitude difference between experiment and theory can also

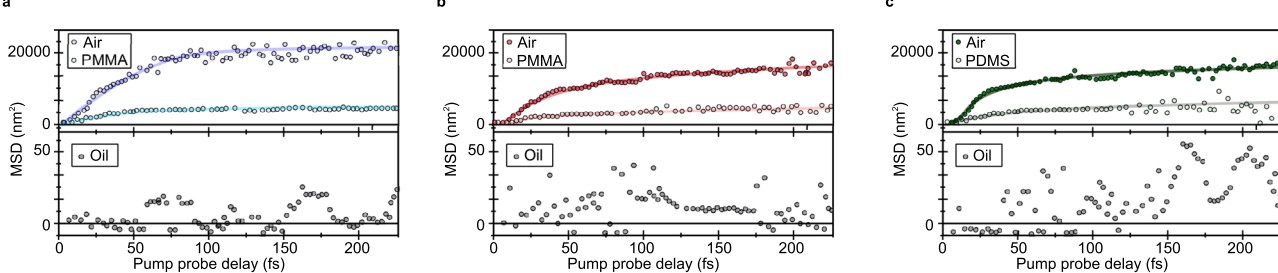

**Fig. 5 Tuning energy propagation in organic nanostructures by refractive index engineering of the external environment. a–c** Mean square displacement of PDA, PDI and PIC samples (static refractive index ~1.55–1.65[79,80]) with an air gap (dark blue, red or green), polymer (light blue, red or green) and oil (black) atop of the sample. The polymer PMMA or PDMS has a refractive index ~1.4 and oil a refractive index of ~1.55; the choice of polymer is guided by sample preparation requirements (see Supplementary Note 14). As the refractive index of the sample and external environment become matched, ultrafast, ultralong transport akin to that observed in Fig. 3 disappears. Solid lines (top panel) are guides to the eye.

not be explained by uncertainties in either (see Supplementary Note 11 for uncertainty calculations). Additionally, we find that in order to obtain a diffusion coefficient comparable to the one measured for the $R_2$ range, the exciton dephasing rate (which is typically used to estimate exciton diffusion coefficients[68] and is related to the homogeneous linewidth) should be on the order of ~10 meV. In PDI, for example, where the experimentally measured diffusion coefficient is lowest, this is already five times smaller than the homogeneous absorption linewidth (~50 meV), hence, transport within the $R_2$ regime is at the limit of what can be described using a pure excitonic model. This is in agreement with previous theoretical calculations which predict exciton diffusion coefficients (and velocities) within the incoherent $R_2$ region between 0.01–20 cm$^2$ s$^{-1}$[62,69,70] (velocities: $0.02 \times 10^6$–$0.08 \times 10^6$ m s$^{-1}$[71–74]). For PDA and PIC our experimental results show diffusion coefficients in $R_2$ well beyond this range and only overlap with the lower end of $D$ values in PDI. Given the presence of EPs determined from our steady-state measurements above, and that especially in $R_1$ but also in $R_2$ the measured transport falls well beyond the range expected for purely excitonic transport, we propose EPs are directly responsible for the vastly enhanced transport velocities observed.

Based on the experimental evidence presented, we conjecture that the two regimes, $R_1$ and $R_2$, detected in our experiments correspond to transport mediated by coherent EPs and by the quasi-localised subradiant states which involve some EP character, respectively. At early times, the excitation will move ballistically according to a group velocity associated with the formation of a coherent EP wavepacket. Rapid population transfer from these EPs into the subradiant states, however, is entropically favourable due to the large density of states of the latter and this transfer should be irreversible given the room-temperature conditions and the small Rabi splittings featured by these systems. The kinetics of this transfer is mediated by vibrational relaxation and should exhibit a similar timescale to that in cavity polariton systems, on the order of ~50 fs[12]. This is also consistent with the timescale of dephasing in the Rabi flopping experiments in Fig. 3. Once population is collected in these subradiant states, transport of energy should be diffusive, yet with an interchromophoric coupling that is mildly enhanced by the admixture with photons due to disorder.

Although the Rabi flopping observed in PDI requires high driving optical fields to achieve a regime where field-dependent intensity effects can be observed, EPs themselves are formed from the interaction with a vacuum field. As such they will always be present and accessible even in the absence of irradiation. This is consistent with our observation of no dependence of the transport behaviour on photon flux (Supplementary Note 9)[75]. We emphasize that because fs-TAM inherently and directly probes

electronic population (unlike e.g. fluorescence based techniques), what we are measuring here is the transport of excitations within the semiconductor material, rather than photonic transport within a waveguide[29,30,44,76] (Supplementary Note 12). This is also in contrast to measurements of 'intrinsic' polaritons in bulk organic (and some inorganic) systems which have been based on detecting the transport of light rather than electronic excitation[17,20,21]. These differences are crucial since specifically transport of excitations is needed for optoelectronic applications. The polaritonic effects observed here, such as long-range energy transport, are also present long (~200 fs) after the removal of a broadband driving optical field. This is in stark contrast to works which have shown EP formation under strict resonant excitation[26,77] (optical field ~0.02 V m$^{-1}$) in macroscopic single crystals with optical densities ~10 times larger than those used here[26,77], and at cryogenic temperatures. To further place the study here in better context it is important to note that Dubin *et al.* demonstrated large transport lengths of up to 10 μm at cryogenic temperatures in the red polymorph of PDA using interferometry[38]. These large migration lengths were initially ascribed to excitonic transport and later suggested to have contributions from a polariton wave propagating in the direction transverse to the chain axis[77,78]. The (ballisitic) transport velocities in this case were however estimated to be one order of magnitude slower[65] than that observed here, closer to that seen in the $R_2$ diffusive phase of the present work. Based on this and the different excitation/material processing conditions, it is possible that at cryogenic temperatures a different mechanism of energy transport operates as compared to that at room temperature which results in slower transport; due to the numerous differences in measurements care must be taken in comparison. We remark that our transport velocities compare to those on standard organic microcavity polaritons at room temperature[10] where velocities of ~$1 \times 10^6$ m s$^{-1}$ were reported. Finally, we note that control measurements on the isolated PIC/PDI dye molecules, unaligned partially polymerised PDA and films of CdSe nanocrystals do not show either $R_1$ or $R_2$ (polaritonic) behaviour (Supplementary Note 13 and hence show that the observations are directly related to the nanostructure of the materials.

To further evidence the mechanism of EP formation and demonstrate the tunability of EP propagation we repeat fs-TAM measurements on all three samples (static refractive index 1.55–1.65[79,80]) with a layer of polymer (PMMA or PDMS, refractive index ~1.4) or index-matching oil (refractive index ~1.55) atop of the samples (Fig. 5, see Supplementary Note 14 for experimental details). As shown by the light circles in Fig. 5a–c, with polymer on top of the organic material the velocity in $R_1$ for all three systems is reduced (PDA: ~$5.0 \times 10^6$ m s$^{-1}$ (air) vs

$1.1 \times 10^6 \, \text{m s}^{-1}$ (polymer); PIC: $\sim 4.1 \times 10^6 \, \text{m s}^{-1}$ (air) vs $\sim 1.1 \times 10^6 \, \text{m s}^{-1}$ (polymer); PDI: $\sim 3.2 \times 10^6 \, \text{m s}^{-1}$ (air) vs $\sim 0.9 \times 10^6 \, \text{m s}^{-1}$ (polymer)). The propagation distance (and diffusion coefficient) in $R_2$ is also reduced for all three materials from 185 nm ($95 \, \text{cm}^2 \, \text{s}^{-1}$) to 76 nm ($14 \, \text{cm}^2 \, \text{s}^{-1}$) (average taken across all three materials). When the refractive index of the organic material and external optical environment are matched with oil no ultrafast EP transport is observed as shown in bottom panels of Fig. 5a–c. This confirms that both $R_1$ and $R_2$ regimes require light-matter coupling, consistent with the models of EP formation and sub-radiant states, and suggest that large Fresnel reflections at the interfaces and the inhomogeneous topography induce optical confinement resulting in strong-coupling effects. These results also create a new methodology by which to control and tune transport behaviour via refractive index engineering. We note the key element of the transport mechanism observed is a refractive index *difference* between the organic film and dielectric environment. Although we have demonstrated this with air, metals (e.g. Ag, refractive index ~ 0.05) or metal oxides (e.g. $MoO_3$, refractive index ~ 2.3), often used at contacts in optoelectronic devices, would provide a similar mismatch.

In summary, we have shown in a range of organic semiconductors, that the initial photoexcitation is not an exciton but an exciton-polariton. These EPs have a defined lifetime, i.e. a precise photonic state energy, coherent lifetime and linewidth that can be observed, and are formed despite the absence of any external cavity structure. We have demonstrated these EPs show an angle-dependent upper polariton/lower polariton dispersion, collective emission and ultrafast, tunable, long-range energy transport, supporting our observations with state-of-the-art theoretical calculations. Our results open the possibility of harnessing cavity QED, typically associated with low temperature physics and epitaxial sample fabrication, in a remarkably simple manner in organic semiconductors and without a cavity. Key requirements for the mechanism of EP formation are: a high material oscillator strength, typically achievable *via* the coupling of ordered dipoles; and a refractive index mismatch between the organic medium and its surroundings. We envisage, in particular, that the transport of electronic excitations using the mechanism showcased here could be used to improve organic optoelectronic devices[12,81].

By tailoring the refractive indices between e.g. metal-oxide charge transport layers and control over film homogeneity to avoid local shorts energy migration lengths could be improved up to five times as compared to the FRET length scales of current devices. This could boost the efficiency of organic photovoltaic cells, by allowing thicker i.e. more absorbing active layers to be used, such that a greater proportion of excitons reach the interfaces/contacts where charge dissociation reaction occurs[60,82]. The absence of an external cavity mitigates issues of charge injection and light in/out-coupling that have plagued other polaritonic devices[83] and could aid in the development of organic exciton/polariton transistors[84]. Furthermore, given our mechanism imposes no constraints on the nature of light, e.g. the use of non-coherent light, such effects could be utilised with sunlight and thin films which are in the range of optical densities studied here (0.2 – 0.6). For low-threshold lasers and LEDs such phenomena could be used to generate emission with narrower linewidths due to the formation of EPs prior to Frenkel excitons[85]. The highly coherent emission observed from PDI nanobelts suggests that if the challenges associated with synthesis and integration of these materials can be overcome they could find use in quantum emitter based devices[86]. The formation of EP states in bare organic films could also enhance chemical reactions such as charge transfer[87] or isomerisation[88] in a manner similar to that demonstrated in organic microcavities[1]. Here, the hybrid light–matter excitation would alter the reaction driving force, with the creation of UP/LP states allowing previously thermodynamically or kinetically unfavourable electronic intermediates to be accessed[12,89]. The ability to tune the strength of a light-matter interaction, *via* the refractive index of the environment, could allow these methods to be applied to materials which are challenge to process in microcavities, e.g. topochemically grown crystals, or when there are light-management constraints. Furthermore, the absence of an external microcavity reduces the device footprint and allows for easier guiding of EPs as recently demonstrated for the Bose-Einstein condensation of polaritons in bare organic crystals[90]. The persistence of cavity-free EP formation at room temperature and in the presence of comparatively high levels of disorder, which would typically hamper many other systems e.g. inorganic quantum wells, suggests a remarkable level of defect tolerance. For example, although GaN polariton lasers have been shown to operate at defect densities of $10^2$ to $10^3 \, \text{cm}^{-2}$ at room temperature and up to $10^8 \, \text{cm}^{-2}$ at cryogenic temperatures[91], above these defect concentrations these epitaxially prepared materials to the best of our knowledge typically no longer can support polariton states. In contrast the relatively inexpensive, solution processed, organic materials used here have defect densities $\sim 10^{16} \, \text{cm}^{-2}$[92] suggesting they may be more suited, if the aforementioned constraints can be overcome, to a range of device conditions. The observations in our work may also support the possibility of polariton-mediated energy transfer in biological photosystems, as recently suggested by several groups[81]. Here, the natural crystal of the biosystems has been proposed to act as a microcavity resulting in strong photon-exciton coupling and enhanced energy transport, akin to the behaviour seen in the film materials here[81,93]. Although we have focussed our attention on quasi-1D systems the results presented are generalisable to any organic system fulfilling the aforementioned design criteria, irrespective of dimensionality and sample morphology. Our results call for the development of a new generation of molecular systems specifically designed to harness cavity-free polaritons.

## Methods

**Femtosecond transient absorption microscopy.** Pulses were delivered by a Yb:KGW amplifier (Pharos, LightConversion, 1030 nm, 5 W, 200 kHz) that seeded two broadband white light (WL) stages. The probe WL was generated in a 3 mm YAG crystal and adjusted to cover the wavelength range from 650–950 nm by a fused-silica prism-based spectral filter. In contrast, the pump WL was generated in a 3 mm sapphire crystal to extend the WL in the high frequency to 500 nm, and a short-pass filtered at 650 nm (Thorlabs, FESH650). The pump pulses were focussed onto the sample using a single-lens oil immersion objective (100×, numerical aperture 1.1 NA) to a diffraction-limited spot of ~270 nm (FWHM, full bandwidth). In contrast, the counter-propagating probe pulses were loosely focused onto the sample by a concave mirror (FWHM ~15 μm). A set of third-order corrected chirped mirrors (pump WL – Layertec, probe WL – Venteon) in combination with a pair of fused silica wedge prisms (Layertec) compressed the pulses to sub-10 fs at the sample. The transmitted probe light was collected by the same objective used to focus the pump pulses and imaged by an EMCCD camera (Qimaging Rolera Thunder, Photonmetrics). Pump light was suppressed by inserting an appropriate long-pass filter in the detection path in front of the camera. Additionally, a bandpass filter (Semrock/Thorlabs) was additionally placed in front of the camera to image at the desired probe wavelength in order to avoid chromatic aberration induced image artefacts. Differential imaging was achieved by modulating the pump beam at 45 Hz by a mechanical chopper. The axial focus position was maintained by an additional auto-focus line based on total internal reflection of a 405 nm continuous wave laser beam.

**Transmission and reflection spectroscopy.** Optical microscopy was performed using a Zeiss Axio.Scope optical microscope in Köhler illumination equipped with a 100× objective (Zeiss EC Epiplan-APOCHROMAT 0.95 HD DIC) coupled to a spectrometer (Avantes HS2048) via an optical fibre (Thorlabs, FC-UV50-2-SR). Five spectra were collected for each sample using an integration time of 10 ms and 20 ms for reflection and transmission measurements, respectively. The reflectance and transmittance were calculated using a silver mirror (ThorLabs, PF10-03-P01) and the glass substrate as references respectively.

**Angle resolved reflectivity**. Angle dependent reflectivity measurements were performed using an Andor Shamrock SR-303i-A triple-grating imaging spectrograph, with a focal length of 0.303 m. The spectra were recorded using a 300 grooves / mm grating blazed at 500 nm. The reflectivity light source was a fibre-coupled 20 W tungsten halogen lamp (Ocean Optics DH-2000-BAL). The angle dependence was measured using a k-space imaging setup. The white light was focussed on to the sample at normal incidence using an Edmund Optics 20X objective with a numerical aperture (NA) = 0.63, with the reflected signal collected through the same optical path using a beam splitter. This light was then focussed into a spectrometer using a final collection lens. An additional Fourier-plane imaging lens positioned before the final lens facilitated the Fourier plane to be imaged into the spectrometer. Here, a dual axis slit was positioned at the focus of the imaging lens (before the final collection lens) which allowed the reflectivity to be spatially filtered, permitting the unwanted real space signal to be rejected.

**Power dependent photoluminescence spectroscopy**. Samples were excited with ~120 fs pulses, ~10 nm full width at half maximum (FWHM), generated from a tunable optical parametric oscillator (OPO) (Spectra Physics Inspire) pumped at 820 nm with a repetition rate of 80 MHz. The power of the pulses was controlled using a variable neutral density filter (ThorLabs) mounted on a rotating stage. The attenuated pulses pass were focused by a microscope objective (Nikon 100x, numerical aperture 0.9) to excite the PDI nanobelts at 590 nm. Emission light was collected by the focusing objective and passes through tunable long pass filters (Fianum). An achromatic doublet lens (focal length f = 60 mm) focuses the filtered light onto a single photon avalanche diode (SPAD) (MPD, jitter time < 50 ps). The output of the SPAD was read out and averaged over 5 seconds for each power step. In the plot shown in Fig. 2 of the main manuscript, we have accounted for the transmission through the optics and the data is a representative measurement from different single nanobelts.

**Sample preparation**

*Polydiacetylene (PDA)*. The 3BCMU (3-methyl-n-butoxy-carbonylmethyl-urethane) diacetylene monomer molecules were synthesised in-house. To prepare ultrathin single crystals a melt-processing method was used. The whole process was systematically carried out under a polarised optical microscope so as to be able to follow and control the sample elaboration: a very small amount of diacetylene powder is placed at one edge of the double-slides assembly; when heating above the melting temperature (~65 °C) the liquid diacetylene fills the empty space by capillary action to form a thin liquid film between the two substrates. Rapid cooling leads to the formation of a highly polycrystalline film. The sample is then heated again to around the melting temperature until the melting of all the crystallites took place. When only a few crystal germs remain the sample is cooled again at a very slow cooling rate (typically < 0.1 °C/mn) to induce the growth of large single monocrystalline domains from the germs. Highly aligned regions of the crystal were selected using a transmission microscope prior to fs-TAM measurements.

*PIC*. To prepare the aggregates, PIC monomer (Sigma Aldrich) was dissolved in methanol by shaking overnight to produce a 10 mM solution. We then mixed 250 μL of monomer solution with 250 μL of a saturated solution of sucrose. We deposited 200 μl of the sugar-monomer mixture onto pre-cleaned glass substrates (acetone, isopropanol, O₂ plasma etc; each 10 mins) and spin coated at 2500 rpm for 2 mins. All spin coating was performed in an inert N₂ environment, to prevent oxidative degradation. All solvents used were purged of O₂ via a freeze-pump-thaw cycle and bubbling of Argon through the solvent. The samples were then dried under vacuum (0.5 atm) for 24 h, to form a uniform amorphous glass.

*PDI*. The PDI monomer was prepared as discussed in the supporting information. Nanobelts were fabricated by self-assembly of CH-PTCDI molecules using a phase transfer. A concentrated solution of CH-PTCDI (0.5 mL, 0.3 mM) was prepared in chloroform in a glass vial. This was then injected to the bottom of a solution of ethanol (~11:3 volume ratio, EtOH: PDI). The solution was allowed to sit in the dark at room temperature for 24 h and nanobelts were formed at the interface of the solvents. After mixing the two solvents, the nanobelts diffused into the whole solution phase and were transferred onto glass coverslips by pipetting. The samples were then allowed to dry in a N₂ glove box before encapsulation and measurement. The solution was not shaken during the self-assembly process to prevent nanobelts with sharp ends forming.

## Data availability

The data underlying all figures in the main text are publicly available at https://doi.org/10.17863/CAM.74815. CCDC 1914225 contains the supplementary crystallographic data for this paper. These data can be obtained free of charge from The Cambridge Crystallographic Data Centre via www.ccdc.cam.ac.uk/structures.

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

## Acknowledgements

We thank the EPSRC (UK) and Winton Program for Physics of Sustainability for financial support. R.P. thanks O. Zadvorna, T. H. Thomas, A. Tanoh, A. Cheminal and W.M. Deacon (Cambridge) for assistance with experiments and useful advice. The authors also thank D. Beljonne (Mons) and S. Pannir-Sivajothi (UCSD) for fruitful discussions. This project has received funding from the European Research Council (ERC) under the European Union's Horizon 2020 research and innovation programme (grant agreements No 670405 and No 758826). The work of G.J, O.D.O. and S.V. was

supported by the European Research Council (ERC-2014-STG H2020 639088). O.S.O acknowledges the support of a Rubicon fellowship from the Netherlands Organisation for Scientific Research. R.C. acknowledges support from Trinity College, University of Cambridge. T.W. acknowledges a grant from the Swedish research council (VR, 2014-06948) as well as financial support from the Knut and Alice Wallenberg Foundation through the project grant 3DEM-NATUR (no. 2012.0112). SMC acknowledges the Henslow Research Fellowship at Girton College, Cambridge. P.A.M thanks the EPSRC for financial support under grant number EP/R025517/1. C.S. acknowledges financial support by the Royal Commission for the Exhibition of 1851. We acknowledge access and support in the use of the electron Physical Science Imaging Centre (EM20527) at the Diamond Light Source. A.W.C., T.B., L.L., R. S. and F. M. acknowledge the CNRS (France) for financial support. A.J.M., R.J. and D.G.L. acknowledge support from EPSRC (UK) grant EP/M025330/1. M. D. and J.Y.-Z. were supported by the US Department of Energy, Office of Science, Basic Energy Sciences, CPIMS Program under Early Career Research Program award DE-SC0019188.

## Author contributions

R.P., R.Y.S.C. and Q.G. prepared the PIC and PDI samples, performed the fs-TAM measurements (with assistance from J.S.) and analysed the data. C.S. assisted with figure preparation. O.S.O. and R.C. performed Rabi flopping and reflectivity measurements and interpreted the data under the supervision of J.J.B. G.J. and O.D.O. performed microscopic reflectivity measurements under the supervision of S.V. J.G. and F.A. synthesised and characterised the PDI monomers. T.B. (Cambridge) performed KPFM-FM measurements and numerical simulations of the fs-TAM data. T.W., D.N.J. and S.M.C. structurally characterised the PDI nanobelts and solved the crystal structure under the supervision of P.A.M. T.B. (Paris), L.L., R. S. and F. M. synthesised the PDA samples. A.J.M., D.L. and R.J. performed transfer matrix simulations and interpreted the data. A.A. devised the model fit to the fs-TAM data. A.M.A performed GWB-BSE calculations. J.Y.Z. and M.D. performed MEEP calculations. S.K.S developed the exciton diffusion model, performed numerical simulations and along with J.Y.Z. interpreted the data. R.H.F. interpreted the data. P.K. designed the fs-TAM experiment. R.P. and A.R. conceived the idea, interpreted the data and wrote the manuscript with input from all authors.

## Competing interests

The authors declare no competing interests.
