## [Peer Review File · Nature Communications]

Microcavity-like exciton-polaritons can be the primary photoexcitation in bare organic semiconductorsEditorial Note: This manuscript has been previously reviewed at another journal that is not operating a transparent peer review scheme. This document only contains reviewer comments and rebuttal letters for versions considered at *Nature Communications*.

REVIEWERS' COMMENTS

Reviewer #1 (Remarks to the Author):

The authors have done a good job in responding to the reviewer queries. They remeasured some things, toned down the more “fanciful/nonsensical” statements (their words) and put their work into a broader context of the literature. I am happy to go ahead with publication. Apologies for the embarrassing confusion between cm and m on behalf of this reviewer.

I just have a few minor things the authors may wish to check on, but do not to see their response.

In the extensive “summary” section (response to point (iii)) they write “e.g. oxide charge transport layers” – I think they mean metal-oxide.

“find use in quantum emitter based devices^{44,88}” – Ref. 44 has nothing to do with “quantum emitters”, let alone devices. After all the revisions, it is probably a good idea to check all of the references again.

The sentence beginning with “The ability to tune” needs fixing: “these methods to be extend to materials which are challenge to process in...” (sic)

Fig. 2 caption reads “solid black like” (sic)

The authors did not respond to my question whether they measured sweeps up and down in laser power. Perhaps they can at least mention this in the SI (Fig. S36). Also, typo in the caption: “performed using a manner to that described” (sic). Why is the last data point cut off in this figure? There is a line protruding from the point at 116 uW, but it goes off the scale.

Many of the references are without volume/page number (or they are wrong, e.g. Ref. 88). This may well get fixed in the main text, but not in the SI (e.g. Refs. 41, 44, etc.)